Evaluating the effectiveness of online courses in international trade using deep learning

Zhang Zhaozhe 1 13273806279@163.com
Andrés Javier De 2
1 School of Economics and Management, Henan Technical Institute , Zhengzhou , China
2 Department of Accounting, Universidad de Oviedo , Oviedo , Spain
Alatas Bilal
Electronic publication date: 2024 Nov 21
Publication date: 2024
Volume: 10
Electronic Location ID: e2509
Received 2024 Jul 1; Accepted 2024 Oct 21
Copyright: © 2024 Zhang and Andrés
Copyright year: 2024
Copyright holder: Zhang and Andrés
License: This is an open access article distributed under the terms of the Creative Commons Attribution License, which permits unrestricted use, distribution, reproduction and adaptation in any medium and for any purpose provided that it is properly attributed. For attribution, the original author(s), title, publication source (PeerJ Computer Science) and either DOI or URL of the article must be cited.
License URL: https://creativecommons.org/licenses/by/4.0/

Keywords: Deep learning, Implementation effect evaluation, Online international trade course

Funding: The authors received no funding for this work.

==============================
The development of the world economy has prompted various countries to pay more attention to the teaching of online international trade courses based on deep learning. In the Internet age, online teaching has become an essential way for people to receive education. To guide the public in obtaining high-quality online teaching resources related to international trade, we propose an evaluation method for the implementation of international trade online courses based on deep learning. Firstly, by analyzing the characteristics of online education in international trade courses, we decompose the evaluation methods of online courses in international trade. Then, using deep learning technology, we propose a fusion method of multimodal evaluation features of online courses in international trade. Finally, we design a classification model to realize the effect evaluation of the course by inputting the fused features. Experiments show that our method can accurately evaluate the effect of international trade online courses, with an accuracy of 78.53%.

Introduction

With the increasingly frequent economic exchanges in the world, the trade exchanges between various countries and regions are gradually increasing. To acquire more knowledge about economic exchanges, the content of online courses on international trade is becoming more and more abundant. Therefore, we propose an implementation effect evaluation method of an online global trade course based on deep learning to improve the online teaching quality of global trade and the education and teaching level of global trade.

An online international trade course is an online learning course provided through a network teaching platform, which mainly involves knowledge and skills related to international trade (Carballo et al., 2022; Silva & Ademilson, 2022). This kind of online course usually adopts technological means such as the Internet and mobile terminals so that learners can learn independently through the network, and at the same time, they can also interact and communicate with teachers and other students. The teaching content covers the basic knowledge of international trade, trade policies and regulations, international trade practice, international trade risk management, cross-border e-commerce and other aspects. Students can choose learning courses independently on the online learning platform according to their own learning needs and interests and flexibly arrange learning time and pace. Learners can freely choose learning content and flexibly arrange learning time and pace according to their learning needs and interests. Learners can interact and communicate with teachers and other students through online platforms to obtain timely teaching feedback and help. International trade online courses are usually jointly developed and taught by industry experts and education experts, and the teaching content is authoritative and practical, which can help learners better master international trade knowledge and skills. International trade online courses usually use multimedia teaching methods, including text, pictures, video, audio and other forms, so that learners can understand and master the course content more vividly and intuitively (Carballo et al., 2022; Chya, 2022).

With the continuous development of online education, course assessment has become an indispensable part of the international trade teaching process. Traditional course evaluation mainly relies on students’ feedback, test scores, and other data, which has problems such as incomplete data collection and low efficiency of analysis. However, the application of deep learning technology can realize more intelligent and refined analysis and judgment in course evaluation (Xu & Zhou, 2020; Kavitha, Mohanavalli & Bharathi, 2018; Harasim, 2000). Deep learning technology can automatically assess students’ performance and understanding in the course by learning historical data and patterns. Deep learning algorithms are used to automatically classify and evaluate the questions and doubts raised by students in the course, which helps teachers better understand students’ learning needs and difficulties. Personalized online course education is established, and more refined evaluation is carried out according to student’s learning habits, interests and other personalized characteristics. By using students’ learning history and behavior data, customized assessment and feedback can be carried out to help students better find and solve their learning problems. In addition, deep learning technology can realize multidimensional analysis and judgment in course evaluation. Deep learning algorithms are used to evaluate students’ language expression, logical thinking, emotional expression, and other aspects comprehensively, which helps teachers understand students’ learning status and needs more comprehensively. In the online course feedback, deep learning technology can help teachers understand students’ learning situation and performance in a timelier manner, monitor and evaluate students’ test results and answer process in real-time, and help teachers find students’ learning problems and difficulties in time. However, the effect assessment of international trade online courses contain a lot of information, such as the teacher’s course preparation, the students’ response in class, the interaction between the teachers and students, and the homework scores after class. This information is multimodal and difficult to evaluate using a single model.

To solve this problem, this article proposes an education and teaching evaluation method based on deep learning, which aims to improve the quality of international trade online courses, improve its education and teaching level, and inspire the design and planning of teaching. Firstly, we construct an international trade online course implementation effect evaluation system to carry out the method proposed in this article and facilitate the development of online teaching evaluation. Then, we propose a feature integration approach for the implementation effect of online courses in international trade in order to understand the various multimodal characteristics of courses fully. Finally, we propose a classification model of course implementation effect to realize the evaluation of international trade online teaching. In addition, we make a theoretical evaluation of the methods and systems in this article to explore the management of education and teaching in universities. The main contributions are as follows: 1) We introduce an integrated approach for multimodal feature fusion modeling in evaluating online international trade courses. This method combines expert review features, course learning outcome features, and student behavior features using encoding and decoding structures, providing a comprehensive depiction of course implementation effects.

2) Viewing the evaluation of online international trade courses as a classification task, we propose an evaluation method leveraging an enhanced Transformer. This method assesses the overall quality of the teaching process, taking into account the course timeline.

3) Our method outperformed other competitive approaches, achieving superior results in accuracy, model efficiency, and other evaluation metrics.

Related works

Research on evaluation methods of online course effect

The overall research on online course evaluation shows the characteristics of wide research scope, deep research depth, and more mature research methods. Wang, Hu & Zhou (2018) proposed that high registration rates and low completion rates are a significant bottleneck of the current development of Massive Open Online Courses (MOOC). Based on the background of big data, this article constructed a set of semantic analysis models (SMA) to study the overall learning situation and the emotional tendency of learners in MOOC platforms and found that the model can effectively identify learners’ emotional tendencies. This model has significant reference value for improving the quality of MOOC courses and significantly increasing the graduation rate of students. Xie (2019) proposed that massive review data is of great significance for distance education quality research. This article attempted to track learner learning log data in the MOOC platform and predicted many dimensions of teaching optimization, which have certain guiding and practical value. Barteit et al. (2020) conducted a study on the quality of online education in the medical field in low-income countries, selected 12,294 participants as survey objects, and obtained many research conclusions through the methods of questionnaire design and pilot study. At the same time, it was also found that the blindness of participants when they were surveyed was low. Most countries still need to take measures to explore more effective and reliable ways of online course evaluation and evaluation. Gómez-Rey et al. (2018) also built a set of comprehensive online teaching evaluation tools based on the online learning environment, linking teachers’ roles, teachers’ performance and students’ demographic characteristics with teaching effectiveness. At the same time, they obtained many research results, including the fact that teachers’ teaching effectiveness is mainly affected by teachers’ roles and classroom performance. Teaching strategies should promote the integration of technology and teaching. Parker et al. (2018) evaluated more than 20 online teaching resources from four dimensions: content, design, interactivity and usability. They found that the average score of online courses was 73 points. In contrast, videos and web resources only accounted for 48 and 62 points, emphasizing the importance of high interactivity for online teaching effect. Calderon & Sood (2020) selected three dimensions of teaching context (accuracy and complexity), interactive communication quality, and meta-learning (reflection on tasks and learning process) to conduct online course quality evaluation research, aiming to break the traditional evaluation strategy mode based on written assignments or face-to-face communication and emphasize the systematic analysis of online teaching quality from multiple aspects.

With the innovation of online education technology, the aforementioned research has not only deepened our understanding of the complexity of online course evaluation but also provided invaluable theoretical support and practical guidance for enhancing the quality and efficiency of online education. Research in this field is further evolving towards refinement, personalization, and intelligence, with its subtopics (such as personalized learning paths and evaluation, emotional intelligence and psychological analysis, multimodal learning research, etc.) becoming increasingly mature.

Course evaluation methods based on deep learning

A deep learning network is extended from a neural network, which has strong feature learning ability. The learned features can restore the characteristics of data and have good visualization and data classification effects. A deep belief network is a kind of neural network. It initializes the deep belief network layer by layer through an unsupervised learning algorithm, and the time complexity of optimizing deep belief network weights and the network size is linear with depth (Zhang et al., 2018). It takes a simple single-layer problem as the starting point to solve the problem of a complex depth layer constructed by a single layer. It reduces the training difficulty of a deep learning network. Deep learning networks are robust and accurate. Therefore, many studies have carried out online teaching quality evaluation methods based on deep learning networks to improve the quality of online teaching. The online teaching environment is special, so teachers need to change teaching methods and keep track of students’ psychological situation and learning status at any time. By creating an online check-in and assessment mechanism, creating a classroom test link and creating an online teacher’s classroom to answer questions, students’ enthusiasm and attention to participate in online classrooms were improved. Arashpour et al. (2023) proposed a hybrid support vector machine and neural network-based teaching optimizer to predict students’ exam scores. Teaching resources are the main methods to assist online teaching. Teaching resources include online course video materials, online teaching course display and online teaching test question bank, which can enhance students’ interest in learning. Safarov et al. (2023) introduced a novel neural network approach that combines synchronous sequences and heterogeneous features, enabling more accurate generation of candidate resources on e-learning platforms where the number of online educational courses and learners is growing exponentially. Chaudhry et al. (2023) utilized chatGPT to tackle various assignment contents in undergraduate degree programs and compared its performance with top-scoring students. Through deep learning algorithms (LeCun, Bengio & Hinton, 2015), such as convolutional neural network (CNN) (Oyedeji, Khan & Erkoyuncu, 2024), long short-term memory (LSTM) (Du, Li & Xie, 2024) and Transformer (Friedman, Wettig & Chen, 2024), the information in various courses is collected and processed, the quality of the teaching process is summarized, and finally, the evaluation results are obtained.

By innovating teaching methods, enriching teaching resources, and fully utilizing deep learning for data analysis and evaluation, we can provide more personalized, efficient, and high-quality educational services for every student.

The implementation effect evaluation method of international trade online course based on deep learning

Online courses in international trade are top-rated nowadays. In order to improve the quality of international trade online teaching, we constructed an implementation effect evaluation system for international trade online courses. By using deep learning technology to collect and analyze each element in the course, the implementation effect of the course can be directly obtained, as shown in Fig. 1.

Figure 1 Online international trade course evaluation system.

The system mainly evaluates the implementation effect of the course through the following aspects: (1) learning behaviour data, (2) case analysis, (3) course learning outcomes evaluation, (4) learner feedback survey, and (5) expert review. The learning behavior of students was recorded through the course platform or learning management system, including learning duration, learning progress, access frequency and so on. Through data analysis, students’ learning habits and learning behaviours can be understood to provide a reference for course improvement. Representative cases are selected for students to analyze and design solutions. By analyzing the students’ solutions, the effect of the course on the improvement of the students’ practical problem-solving ability was evaluated. At the end of the course, the learning outcomes of the students were evaluated through the course quizzes or examinations. The assessment results can be used to measure the teaching effectiveness of the course and the mastery level of the learners. Feedback on the course was collected through questionnaires or questionnaires. The contents of the survey included course content, teaching methods, learning experience, and learning outcomes. By analyzing the feedback data, the teaching effect of the course was evaluated, and the learning needs of the students were understood to provide a reference for the improvement of the course. Experts were invited to review the course to evaluate the accuracy, authority and practicability of the course content. Through the expert review, the teaching level and teaching quality of the course were determined to provide a reference for course improvement.

Feature integration of the implementation effect of online international trade course

There may be various disturbances in the process of integrating the teaching effect features of international trade online courses. For the three features in Fig. 1, the most common interference is the transformation deviation between modalities, which will not only make the teaching effect features challenging to identify in semantics but also lead to the evaluation error of the teaching effect. Therefore, it is necessary to concatenate and saliency these three types of teaching effect features to ensure that the fused teaching effect features can present clear semantic information.

According to the above characteristics, we constructed a feature fusion network for the implementation effect of online courses of international trade based on deep learning. The deep learning network was trained in a supervised manner by using the processed feature pairs of non-transformation and transformation teaching effects. The feature fusion network’s implementation effect is shown in Fig. 2. The first two stages of the network are based on the encoder-decoder sub-network to extract multiscale context features. In the last stage, the features of the course implementation effect were manipulated to generate the spatial connection output. Between each stage, a supervised attention module was added. Under the supervision of the non-transformation-transformation teaching effect feature pair, the module transformed the feature map of the previous stage and passed it to the next stage. A cross-stage feature fusion mechanism is introduced, and the multiscale context features of the early sub-network are used to consolidate the intermediate features of the later sub-network. In the training process, the loss function is used to calculate the loss of the three stages in a loop to optimize the network model. Finally, the text image with motion blur is input into the trained network model, and the clear image after deblurring is output.

Figure 2 The fusion model of assessment features for online international trade courses.

The encoder-decoder-based sub-network is composed of a Transformer Block, which contains a multi-head self-attention module. Each module is composed of layer normalization (LN), multi-head self-attention (MSA) mechanism, residual connection and multi-layer perceptron (MLP). The calculation process is present in Eqs. (1), (2).

(1) z^i=MSA(LN(zi−1))+zi−1

(2) zi=MLP(LN(z^i))+z^i

where zi-1 denotes the input feature of online courses, and MSA denotes the multi-head attention module. Each MSA is composed of a self-attention mechanism, and its calculation process can be shown in Eqs. (3)–(6).

(3) Q=FWQ

(4) K=FWK

(5) V=FWV

(6) Z=Softmax(QKTdk)V

where WQ,WK,WV refer to the weight matrixes, F represents each modal feature, and Z denotes the fused multimodal feature. Instead of using a normal CNN or LSTM, we use a subnetwork based on encoders and decoders. This is because we use the concatenation approach when dealing with multimodal features. The subnetwork based on an encoder and decoder can fully explore the internal relationship between various features. At the same time, CNN or LSTM are not suitable for the establishment of the internal relationship of such hybrid features.

Evaluation model of implementation effect of international trade online course

After obtaining the fused multimodal features, we propose a quality evaluation model of international trade online courses based on fused features to evaluate the teaching effect of international trade online courses.

Considering the multimodal features we extracted for the course, we chose the bidirectional encoder representations from Transformers (BERT) model as our classifier to get the evaluation results. Considering that BERT is a pre-trained model, the input sequence of the model must be the multimodal fusion features completed by concatenation. This further reflects the importance of the above feature fusion. The input of BERT is the representation corresponding to each token, and the feature military of different modalities is constructed by the WordPiece algorithm. The representation is composed of three parts, namely the corresponding token, segmentation and position embeddings. To accomplish the specific classification task, in addition to the token of the word, the authors also insert a particular token of classification at the beginning of each sequence of the input, and the final output of the Transformer layer corresponding to the classification token is used to aggregate the representation information of the whole sequence. As shown in Fig. 3, the basic unit of the Bert model is the Transformer module, so its basic principle is shown in Eqs. (1)–(6).

Figure 3 The fusion model of assessment features for online international trade courses.

In the BERT model, we calculate the correlation between the information at different times of the course. Unlike the encoder-decoder structure in the previous section, the Transformer module here is oriented to details about a time axis. Although the basic unit is the same, the objects and ways of dealing with them are entirely different. Using the features of an online course on international trade at other times, we fully calculate the interaction between similar features based on the timeline to obtain the comprehensive features of the course over the whole period. Through these extensive features, we can get the results of a thorough evaluation of current international trade courses.

The use of deep learning technology in the implementation effect assessment of international trade courses

Through the deep learning technology, the establishment of the implementation effect evaluation model of international trade online courses is conducive to us to re-examine the development and education activities of international trade online courses, as shown in Fig. 4.

Figure 4 The effectiveness of online international trade curriculum.

Using the method mentioned above can effectively improve the efficiency and accuracy of curriculum evaluation. We improve the efficiency and accuracy of assessment by building automated assessment tools and algorithm models. Natural language processing technology is used to analyze the articles and reports submitted by students and assess their quality and accuracy automatically, reducing the assessment burden of teachers. In addition, deep learning technology can help analyze students’ learning behaviours and outcomes. Students’ engagement and learning outcomes are evaluated by analyzing their interaction, browsing and discussion records in online classes. The results of assignments, essays, and tests submitted by students were analyzed and evaluated to assess their learning outcomes and quality. According to the characteristics of deep learning models, the quality of international trade courses was evaluated and monitored in a big data-driven manner. According to the learning situation and performance of students, the course content and teaching strategies were updated in real-time to improve the teaching effect and attraction of the course. At the same time, by monitoring students’ learning outcomes and learning behaviors, the course design and teaching management are optimized, and the overall quality and teaching level of the course is improved. Taking into account the above factors, deep learning technology can help course evaluators establish multidimensional evaluation indicators and evaluate the quality of international trade courses from different perspectives. Combined with student feedback and satisfaction survey, the teaching effect and satisfaction of the course were evaluated. At the same time, the overall quality and level of the course were assessed by combining the knowledge point coverage, teaching quality and course design. In addition, relying on the powerful computing power of the hardware, we can personalize students’ learning plans based on students’ learning situations and needs through deep learning and provide personalized evaluation and teaching services. According to students’ learning progress and mastery level, we can adaptively adjust the difficulty and method of assessment and provide personalized assessment guidance and suggestions. At the same time, the recommendation system of deep learning technology is used to recommend relevant learning resources and learning methods.

Experiment and analysis

Dataset and implement details

We use Udemy Courses as a dataset (https://zenodo.org/records/11202648, doi: 10.5281/zenodo.11202648) and a COIN dataset (https://coin-dataset.github.io/, doi: 10.1109/CVPR.2019.00130) to verify the effectiveness of our method. Udemy Courses dataset comprises 3,682 records, each detailing courses from four distinct subject areas: Business Finance, Graphic Design, Musical Instruments, and Web Design. These courses are sourced from Udemy, a renowned MOOC platform that offers a diverse range of educational content, both at no cost and for a fee. We extracted the relevant parts about international trade courses from the dataset and made statistics. The relevant information on international trade courses in the dataset is shown in Table 1.

Table 1 Statistics of international trade course data.

COIN dataset and Udemy Courses dataset	Number	
Expert assessment score	2,234	
Learning behavior information	6,324	
Achievements checking information	9,441	

The experiment is carried out on a device with i5-13500 Cpu and Rtx 3090 Gpu, the operating system is RedHat, and the network model is implemented under the MxNet framework. The total number of rounds of the experiment was set to 120, the batch size was set to 64, and the initial learning rate was set to 0.01. Adam was used as the optimizer of the model, the momentum was 0.9, and the weight decay term was set to 1×10−5.

In order to evaluate the performance of the model, we use accuracy as the evaluation criterion, which is calculated in Eq. (7):

(7) Accuracy=TP+TNTP+TN+FP+FN

where TP refers to the samples which are correctly classified, TN represents samples that are actually negative and predicted to be negative, FP denotes the incorrectly classified samples, and FN denotes samples that are actually positive but predicted to be negative.

Results and discussion

We evaluate the effectiveness of an implementation method for assessing online international trade courses using deep learning on a curated dataset of such courses. Various models adept at processing multimodal features are chosen, including CNN-LSTM (Aksan et al., 2023), Transformer (Friedman, Wettig & Chen, 2024), CNN-Transformer (Li et al., 2023), Dlinear (Zeng et al., 2023), ViT (Han et al., 2022), Deit (Touvron et al., 2021), Clip (Conde & Turgutlu, 2021), and Informer (Zhou et al., 2021), with their results compared in Table 2. Notably, our method achieves the highest performance score, ranking first with 78.53% accuracy. Specifically, it surpass CNN-LSTM and CNN-Transformer by 13.30% and 7.30% in accuracy, respectively, and outperform the latest Clip and Informer methods by 1.64% and 0.86%, respectively. This is because we have fully integrated the features, which enables us to process online course information more comprehensively. At the same time, we fully consider the course from a temporal perspective, unlike these comparative methods that directly classify it into a single category. However, it is evident that our method falls in the middle range in terms of model parameter count and running time, with values of 78.85 million parameters and 401 s, respectively. While ensuring model performance, our model possesses a relatively high number of parameters and a longer inference time. Yet, considering that CNN-based models rapidly lose multimodal feature details, while LSTM and transformer-based models miss some global features and face classification inaccuracy, our model struck a balance. By judiciously simplifying its structure while prioritizing accuracy, it ensures sensitivity and prevents feature loss.

Table 2 Performance of our method and other methods.

	Acc	Parameters	Time (ms)	
CNN-LSTM	65.32	67.78	328	
Transformer	66.33	71.28	356	
CNN-Transformer	71.23	77.89	398	
Dlinear	74.65	66.89	353	
Vit	75.53	78.9	389	
Deit	74.26	79.34	465	
Clip	76.89	76.54	424	
Informer	77.67	79.54	412	
Ours	78.53	78.85	401	

Finally, we show the model convergence when our method, Clip, and Informer are trained, as shown in Fig. 5. It can be found that the implementation effect evaluation method of the international trade online course based on deep learning proposed by us can complete the training goal of the model faster and reach the state of model parameter convergence in the training process. In addition, we also show the classification results of our method, Clip and Informer, when three different features are used to evaluate the course effect separately, as shown in Fig. 6.

Figure 5 The training of our method, Clip and Informer.

Figure 6 The results of our method, Clip and Informer.

Application test

In order to verify the application of the system proposed in this article in real scenarios, we recruited four-course experts and 30 volunteers to carry out four courses of online teaching of international trade in 4 weeks, simulating the status of the course. Then, our system was used to evaluate the simulated course, and the final course experimental results were obtained from the expert evaluation, the knowledge obtained by the learners, and the class status of the learners. The results are shown in Table 2, and we randomly selected a part of them for display, where 0, 1, 2, 3, and 4 represent the five grades of perfect, excellent, good, medium, and bad, respectively. In addition, we also found four experts in international trade courses to give objective marks to the four classes as our experimental standards.

We can find that in Table 3, for lesson No. 1, our system gives out a good rating, and half of the volunteers’ evaluations are the same as the result of our system. For lesson No. 2, the vast majority of the evaluation results are intermediate, which is the same as the results of our system. For lesson No. 3 and lesson No. 4, our systematic evaluation ratings are perfect and medium, respectively, the same as the results of the comprehensive evaluation. For each online learning, our system can accurately classify the mainstream effect level of the course, providing good technical support for improving the level of international trade online teaching.

Table 3 Test of our application.

Volunteer	1	2	3	4	5	6	7	8	9	10	System	
No. 1 lesson	4	2	1	0	2	2	3	2	0	2	2	
No. 2 lesson	2	3	3	3	1	4	0	3	3	3	3	
No. 3 lesson	0	0	1	2	0	1	0	2	0	0	0	
No. 4 lesson	3	2	3	1	3	3	1	3	3	3	3	

Discussion

Through the aforementioned experiments, we have convincingly established the validity and reliability of the feature fusion method and accompanying evaluation model for assessing the implementation effectiveness of international trade online courses. This verification process not only delves into the rationality of the system design from a scientific theoretical perspective but also showcases the system’s superior performance and practicality in enhancing the implementation outcomes of international trade online courses, providing a solid foundation for further optimizing and refining the online education evaluation system.

Furthermore, we have discovered that the evaluation system, through its refined feature fusion strategy, is capable of comprehensively and accurately capturing multidimensional factors that influence the effectiveness of international trade online courses. These factors encompass students’ learning behaviour patterns, the compatibility between course content and teaching methodologies, and the efficiency of interactive feedback mechanisms. This profound analytical capability enables the system to offer specific and actionable improvement suggestions to educators and curriculum designers, facilitating the precise adjustment of teaching strategies, optimization of course content, and, ultimately, enhancing teaching quality and learning outcomes. In the application, the system’s real-time feedback mechanism significantly improves the flexibility and responsiveness of the educational process. It empowers educators to swiftly identify issues and challenges faced by students during their learning journey, enabling them to adjust teaching plans or provide personalized learning support accordingly. This student-centred teaching model fosters active participation and autonomous learning, effectively boosting students’ learning satisfaction and sense of accomplishment.

Ultimately, we aspire to drive the comprehensive advancement and development of international trade online education through the widespread adoption of this evaluation system, contributing to the cultivation of professionals with global perspectives and competitiveness.

Conclusion

To achieve a precise assessment of the educational outcomes of international trade online courses, we introduce a deep learning-driven approach explicitly tailored to evaluate their implementation effectiveness. By meticulously examining the unique attributes of international trade online courses and harnessing the power of deep learning technologies, we devise a feature fusion strategy exclusively for these courses. Furthermore, we designed an evaluation-level recognition model that enables the accurate categorization of the teaching effectiveness of online international trade courses. Our experimental findings underscore the capability of this method in delivering precise evaluations, thereby furnishing technical underpinnings for enhancing international trade education. In the future, we will explore how to fuse more features, such as teaching content, examination, and student reaction, which can represent the quality of courses. Meanwhile, we plan to research cross-modal methods to boost the evaluation of teaching online using these novel features.

Supplemental Information

Supplemental Information 1 Code.

Additional Information and Declarations

Competing Interests

Author Contributions

Data Availability

The authors declare that they have no competing interests.

Zhaozhe Zhang conceived and designed the experiments, performed the experiments, analyzed the data, performed the computation work, prepared figures and/or tables, authored or reviewed drafts of the article, and approved the final draft.

Javier De Andrés conceived and designed the experiments, performed the experiments, analyzed the data, performed the computation work, prepared figures and/or tables, authored or reviewed drafts of the article, and approved the final draft.

The following information was supplied regarding data availability:

The code is available in the Supplemental File.

The third-party Udemy Courses dataset is available at Kaggle and Zenodo:

- https://www.kaggle.com/datasets/andrewmvd/udemy-courses.

- None. (2024). Udemy Courses [Data set]. Zenodo. https://doi.org/10.5281/zenodo.11202648.

The third-party COIN dataset is available at: https://coin-dataset.github.io. Referenced at doi: 10.1109/CVPR.2019.00130.

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
