# Peer review of "Evaluating the effectiveness of online courses in international trade using deep learning"

_PeerJ Computer Science, doi:10.7717/peerj-cs.2509_

## Round 0.1 · original submission · Major Revisions

· Academic Editor

Major Revisions

Dear authors,

Thank you for the submission. The reviewers’ comments are now available. It is not suggested that your article be published in its current format. We do, however, advise you to revise the paper in light of the reviewers’ comments and concerns before resubmitting it. The followings should also be addressed:

1. Keywords should be written following a unique style. They should also be listed alphabetically.
2. Originality, novelty, and motivation should be clearly mentioned.
3. There is not a clear categorization of related work. Introduction section seems broad, voluminous, and heterogeneous Authoritative synthesis assessing the current state-of-the-art is absent. You should focus on the main topic of the study and present a literature review in tabular form to make it easy to identify research gaps and innovations.
4. Many of the equations are part of the related sentences. Attention is needed for correct sentence formation.
5. Equations should be used with correct equation number. Please do not use “as follows”, “given as”, etc. Explanation of the equations should also be checked. All variables should be written in italic as in the equations. Their definitions and boundaries should be defined. Necessary references should be provided.
6. All of the values for the parameters of all algorithms selected for comparison should be given.
7. Pay special attention to the usage of abbreviations. Spell out the full term at its first mention, indicate its abbreviation in parenthesis and use the abbreviation from then on. See for example MOOC in the paper.
8. Some more recommendations and conclusions should be discussed about the paper considering the experimental results. The conclusion section is weak. There is also no discussion section about the results. It should briefly describe the results of the study and some more directions for further research. You should describe the academic implications, main findings, shortcomings and directions for future research in the conclusion section. The conclusion in its current form is generally confused. What will be happen next? What we supposed to expect from the future papers? So rewrite it and consider the following comments:
- Highlight your analysis and reflect only the important points for the whole paper.
- Mention the benefits
- Mention the implication in the last of this section.

Best wishes,

Reviewer 1 ·

Basic reporting

The paper is not well and lacks many important details to clarify the problem and the proposed method. The authors should have clearly stated and defined the problem and the contribution in relation to the field.

Experimental design

The proposed method itself is not well-defined and appears overly simplistic given the complexity of the problem at hand. The description of the methodology lacks the necessary depth and detail to allow for replication or thorough understanding by other researchers in the field.

Equation number 7 is not the correct formula for calculating the accuracy.

Validity of the findings

The author should provide details of the dataset used in the study and the features.
There should be a discussion section to thoroughly analyze the results.

Additional comments

no comment

Reviewer 2 ·

Basic reporting

Clear and unambiguous English language is used throughout the paper. The Introduction section describes the research question appropriately, criticizes some traditional approaches to course evaluation, and states that the use of deep learning technology can perform better than traditional approaches to course evaluation. Finally, it presents the research objectives.

The Related Work section outlines the knowledge gap that the authors want to fill. The Introduction adequately introduces the topic and makes clear what the motivation is. The authors cited only 14 well-referenced papers, but half of them are from 2018 or 2019. Five papers are written in Chinese, which limits their impact. There is no paper from 2024.

The paper structure is adequate.

Experimental design

The content of this article is not within the scope of PeerJ, which considers articles in Biological Sciences, Environmental Sciences, Medical Sciences, and Health Sciences. The content of this article focuses on the use of data mining techniques to improve the performance of online training courses on International Trade. Therefore, this topic does not have clear applicability to the core areas of Biological, Environmental, Medical, or Health Sciences.

Validity of the findings

No comment.

Additional comments

No comment.

Reviewer 3 ·

Basic reporting

All comments have been added in detail to the last section.

Experimental design

All comments have been added in detail to the last section.

Validity of the findings

All comments have been added in detail to the last section.

Additional comments

Review Report for PeerJ Computer Science
(Implementation effect evaluation and teaching Enlightenment of Online International Trade Course based on Deep learning)

1. Within the scope of the study, an analysis was made using deep learning in relation to international trade online courses.

2. The importance of the subject is clearly and explicitly stated in the introduction section. However, the differences of the study from the literature and its main contributions to the literature should be added in detail at the end of this section.

3. The related works section is addressed from two perspectives and definitely needs to be detailed. Here, it is recommended to add a literature table, especially regarding the deep learning models and results used in the literature.

4. The system in Figure-1, the fusion model in Figure-2 and 3 are clearly stated and explained in detail.

5. When the results are examined in terms of the proposed model and the compared models, it is observed that only accuracy is obtained. In addition, although the use of these models, most of which are transformer-based, is important for comparison, it should be explained in more detail why these are preferred, despite the fact that there are many different models in the literature regarding the solution of this problem.

As a result, although the subject discussed is interesting, attention should be paid to the sections written in bullet points above in order to fully reveal its contribution to the literature.

·

Basic reporting

1. The publication of the authors corresponds to the theme of the journal, professional English was used in the text in an understandable form and with appropriate terminology.
2. The introduction to the publication contains the sentence "Experiments show that our method can accurately estimate the effect of online international trade rates with an accuracy of 78.53%, which is the highest level in the world" - a proposal to delete the words "which is the highest level in the world", because the authors do not prove this fact in a publication at world levels (without analysis and ranking as confirmation).
3. There is a need to expand the literature, much of which is in Chinese, and includes a significant number of studies available in the scientometric databases Scopus, Web of Science and others.
4. The introduction adequately presents the topic and explains the motivation of the research.

Experimental design

1. The methods are described in sufficient detail.
2. The evaluation indicators and the selection method model are presented in the form of a comprehensive study.
3. All abbreviations in the formulas should be clarified, there are gaps in detail (formulas 1-6).
4. Chapter 2 Related works requires a more critical and comparative analysis of sources and a presentation of the evolution of changes in views in research.

Validity of the findings

In the conclusions, it is expedient to present future directions of research where these developments can be used

Additional comments

The publication makes a positive impression regarding the potential interest among professionals and practitioners in this field, is relevant in content and timely.
Minor revisions are needed in the article that do not reduce the value of the conducted research:
1. Expansion of literary sources by geographical feature and coverage of the analysis of a larger area of ​​available scientific research
2. Detailing of the initial data of the experiment
3. Providing interpretation to abbreviations in formulas where it is omitted.

---

## Round 0.2 · Minor Revisions

· Academic Editor

Minor Revisions

Dear authors,

Thank you for the revision. Two of the previous reviewers did not respond to review your paper. Although other two reviewers recommend that your paper be accepted, it seems that you did not clearly address the editor's and other one reviewer's comments an critisisms. You have not performed necessray additions and modifications raised by editor as listed below:

- Keywords should be written following a unique style. They should also be listed alphabetically.
- Authoritative synthesis assessing the current state-of-the-art is absent. You should focus on the main topic of the study and present a literature review in tabular form to make it easy to identify research gaps and innovations.
- Originality, novelty, and motivation should be clearly mentioned.
- Many of the equations are part of the related sentences. Attention is needed for correct sentence formation.
- Pay special attention to the usage of abbreviations. Spell out the full term at its first mention, indicate its abbreviation in parenthesis and use the abbreviation from then on. What is CLS? "... a renowned MOOC (Massive Open Online Courses) ..." and etc. are not corrected.

We encourage you to address these clearly and appropriately, along with the concerns and criticisms of the previous Reviewer1, and resubmit your article once you have updated it accordingly.

Best wishes,

Reviewer 3 ·

Basic reporting

All comments have been added in detail to the last section.

Experimental design

All comments have been added in detail to the last section.

Validity of the findings

All comments have been added in detail to the last section.

Additional comments

Review Report for PeerJ Computer Science
(Implementation effect evaluation and teaching Enlightenment of Online International Trade Course based on Deep learning)

When the changes made in the revised paper are examined in detail, it is observed that they are at the appropriate level. For this reason, I recommend that the paper be accepted. Best regards.

·

Basic reporting

No comment

Experimental design

No comment

Validity of the findings

No comment

Additional comments

No comment

---

## Round 0.3 · accepted · Accept

· Academic Editor

Accept

Dear authors,

Thank you for clearly addressing all the concerns and criticisms. It seems that all necessary additions and modifications are performed. Your paper is acceptable after this last revision.

Best wishes,